# Clinical profile of HIV-infected adults receiving a holistic approach of care model in Nakawa, Kampala District

**Emmanuel Sendaula** [1] *, **Michael Buwembo** [1], **Denis Opio** [2]

**1** Reach Out Mbuya Community Health Initiative, Kampala, Uganda, **2** Epicentre-MSF, Paris, France

☯ These authors contributed equally to this work.
* esendaula@reachoutmbuya.org

**Data Availability Statement:** All relevant data are within the paper and its Supporting Information files.

**Funding:** the author(s) received no specific funding for this work.

## Abstract

People Living with HIV (PLHIV) are often dealing with a range of issues that make life more difficult because of the limited emotional, spiritual, psychological, social, physical and clinical support which consequently lead to poor physical health and quality of life. The holistic care of individuals infected with HIV/AIDS involves promoting psychological and physiological well-being as well as fostering socio-cultural relationships and supporting the fulfillment of spiritual aspirations. We conducted a retrospective cross-sectional study among HIV-infected patients receiving a holistic approach of care model from January 2015 to December 2018 in Kampala district, Uganda. The study involved adult individuals aged 18 and above from whom demographics and other information were obtained. All eligible participants were selected using stratified random sampling from the parishes and systematic random sampling to select study participants. We investigated the clinical profile and the factors associated with viral load suppression among HIV-infected patients receiving a holistic approach of care model in Kampala District. The data was analyzed using STATA version 13. 0. Results: A total of 910 patients were enrolled. 676 (74.3%) were female; 453 (49.8%) were between 18 and 39 years. 324 (35.6%) were either overweight or obese. 769 (84.5%) had viral load beyond detectable limits, 904 (99.3%) were adhering to HIV treatment. 867 (95.3%) were virally suppressed. The age group 40–59 years (Adjusted Odds Ratio (aOR) = 2.85, 95% Confidence Interval (CI):1.36–5.97, P = 0.005) and good adherence (aOR = 12.9, 95%CI:1.86–81.07, P = 0.009) were significantly associated with viral load suppression. Conclusion: The holistic care model supports patients in all facets of their lives, resulting into improved treatment outcomes. Our findings show that age and adherence are linked to viral load suppression among HIV-infected adults receiving a holistic approach of care model.

## Introduction

HIV/AIDS continues to be one of the world's major public health issues, with Sub-Saharan Africa being the most affected region [1]. In Uganda, the HIV prevalence among adults aged 15–64 years is 5.5% [2]. The UNAIDS 2019 data report shows gains in the treatment of HIV-

**Competing interests:** The authors have declared that no competing interests exist.

positive persons. For instance, the annual number of global deaths from AIDS-related illness among people living with HIV (all ages) has declined from a peak of 1.7 million [1.3–2.4 million] in 2004 to 770 000 [570 000–1 100 000] in 2018. Since 2010, AIDS-related mortality has declined by 33%. [3]. Uganda has been reported as the most successful country in Africa in reducing the prevalence of HIV/AIDS from 18% to 6.4% over the last two decades [4]. Available evidence indicated that despite such a significant decline, HIV prevalence has stagnated over the last 5–9 years, at between 6.1 and 6.5% and it is rising in some parts of the country. This is thought to be amassed by the social-psychological well-being of people living with HIV/AIDS [4,5]. It's also noted that only 78% of people on antiretroviral therapy (ART) in Uganda have achieved viral suppression [6].The primary goal of Highly Active Antiretroviral Therapy (HAART) in HIV-infected individuals is sustained viral load (VL) suppression which in turn leads to prevention of disease progression, transmission and improved treatment outcomes [7]. Increasing survival of people living with HIV (PLHIV) must involve strategies that enhance viral suppression. At the peak of HIV/AIDS infection in Uganda, the Catholic Bishops invited people to commit to solidarity with victims of HIV/AIDS and their families [8]. The call entailed engagement in a variety of ministries to aid the victims and their families. Among those who responded to the solidarity call by the Catholic Bishops is Reach Out Mbuya Community Health Initiative (ROM).ROM is a faith-based Non-Government Organization (NGO) with a mission "to curb the further spread of HIV infection among the less privileged members of society in communities and enable those already living with HIV and AIDS to live a responsible and dignified life through a holistic model that entails physical, emotional, social and spiritual needs of the community to restore hope to the hopeless and good health to the sick [9]. Available literature showed that holistic care is derived from the philosophy of Holism, which is the understanding of people by addressing factors that affect them in all situations [10] and is based on humanism, unity emphasizing that the whole is greater than the sum of its parts, such as a person's mind and spirit affect [11]. Holistic care is a comprehensive people-centered model [12] and is described as a behavior that recognizes a person as a whole and acknowledges the interdependence among their biological, social, psychological, and spiritual aspects. Providers of holistic care consider a patient as a whole within her/his environment and believe that a patient is made up of body, mind, and spirit [13]. Without a holistic approach to HIV, the global community risks failing another generation to HIV/AIDS [9]. In Uganda, it is envisioned that people-centered systems of care will improve the quality of services offered to clients, and maximize the efficiency and cost-effectiveness of the country's ART program [14]. Previous studies have highlighted the importance of the holistic model of care approach to the people living with HIV [7,15,16]. However, there is limited information about the performance of this approach to the health outcomes of people living with HIV. we set out to describe the clinical profile and the factors associated with viral suppression status among HIV-infected patients receiving a holistic approach of care model in Kampala District.

## Methods

We conducted a retrospective cross-sectional study among HIV-infected patients who were receiving a holistic approach of care model from January 2015 to December 2018. Data was abstracted from Reach Out Mbuya HIV/AIDS Initiative (ROM) facilities in Kampala district, Uganda. There were 6183 active clients from 2015 to 2018. Patient records were obtained from ART registers that were updated during patient visits to the facilities. Additionally, data is also stored on Open Medical Record System (Open MRS), an HIV data management computer program that helps to digitize patient information. Participants in the study were adult individuals aged 18 and above, on ART and receiving a holistic approach of care model in Nakawa

division Kampala. Using Kish Leslie 1965 sample size formula and a design effect of 2, the required sample size was 910. Participants were selected using stratified random sampling from the Nakawa division parishes and study participants were selected using systematic sampling. Simple random sampling was used to select the first participant in the database, and thereafter, every second participant was selected.

Descriptive statistics were used to describe the study participants. The binary logistic regression was used to determine the factors associated with viral load suppression. The factors with a p-value less than 0.2 were considered for multivariate analysis. Interaction and confounding were assessed in the regression model. The goodness-of-fit test was done on the final model. The Odds Ratios were used as the measure of association and p-values less than 0.05 suggested statistical significance.

## Ethical consideration

We obtained ethical approval from the Clarke International University-Research Ethics Committee (CIUREC/0169) and the Uganda National Council of Science and Technology (Ref: HS903ES). To ensure patient protection and confidentiality, patient identifiers were eliminated by the use of serial numbers.

## Results

### Description of the study participants

The study was conducted among 910 patients receiving a holistic care model in the Nakawa Division, Kampala. A quarter of them were male (25.7%), and almost half (49.8%) were young adults. Of the 676 female patients, 152 (22.5%) were on the Prevention of mother-to-child transmission (PMTCT) program. (See Table 1).

More than a third of the patients (35.6%) were either overweight or obese. Only 141 out of the 910 patients (15.5%) had a detectable viral load. Many of the patients (65.8%) were on Tenofovir-Lamivudine-Efavirenz (TDF-3TC-EFV) HIV regimen. Only 6 of 910 patients did not have good adherence. (See Table 2 for details).

### Factors associated with viral load suppression

At bivariate analysis, being a middle-aged adult (40–59 years) compared to a young adult (18–39 years) was significantly associated with viral load suppression. Other socio-demographic factors- sex, being on the PMTCT program were not associated with load suppression (See Table 3).

Good adherence was significantly associated with viral load suppression whist BMI, and HIV regimen were not associated with viral load suppression at the bivariate level (Table 4).

**Table 1. Socio-demographic characteristics of 910 HIV- infected patients receiving a holistic approach of care model in Nakawa division, Kampala.**

| Variable | Category | Frequency | % |
|---|---|---|---|
| **Sex** | Female | 676 | 74.3 |
| | Male | 234 | 25.7 |
| **Age** | 18–39 years | 453 | 49.8 |
| | 40–59 years | 417 | 45.8 |
| | 60 years and above | 40 | 4.4 |
| **Prevention of Mother-to-Child Transmission (PMTCT)** | No | 524 | 77.5 |
| | Yes | 152 | 22.5 |

**Table 2. Clinical characteristics of HIV-infected patients receiving a holistic approach of care model in Nakawa, Kampala.**

| Variable | Category | Frequency | % |
|---|---|---|---|
| **Body Mass Index (BMI)** | Underweight | 67 | 7.4 |
| | Normal weight | 519 | 57.0 |
| | Overweight | 219 | 24.1 |
| | Obese | 105 | 11.5 |
| **Viral load Detection** | Beyond detectable limit | 769 | 84.5 |
| | Detected | 141 | 15.5 |
| **HIV regimen** | TDF-3TC-EFV | 599 | 65.8 |
| | AZT-3TC-NVP | 171 | 18.8 |
| | Other regimens | 140 | 15.4 |
| **Adherence** | Good | 904 | 99.3 |
| | Not good | 6 | 0.7 |
| **HIV Viral Load** | Suppression | 867 | 95.3 |
| | Non-suppression | 43 | 4.7 |

The adjusted logistic regression model suggested that factors- being in age group 40–59 years compared to being 18–39 years (Adjusted Odds Ratio (aOR) = 2.85, 95% Confidence Interval (CI): 1.36–5.97, P-value = 0.005), and good adherence (aOR = 12.29, 95%CI: 1.86–81.07, P-value = 0.009) were significantly associated with viral load suppression. Whereas BMI was not associated with viral load suppression (See Table 5).

## Discussion

### Clinical profile

The holistic approach requires consistent care and support to facilitate immediate access to treatment when a person is diagnosed with HIV and promotes adherence to treatment for people living with HIV to attain Viral Load suppression. [15]. It's no surprise then, that the vast majority of patients (95.3%) showed that they had suppressed viral load higher than the third 90 UNAIDS strategy. Since the available model of care aims to relink patients that have dropped out of care, it is unquestionably beneficial to medication adherence. In our sample, virtually all patients (99.3%) had good adherence to their antiretroviral therapy, with just 0.7% having poor adherence. Thus, having a holistic model of treatment contributes to higher viral load suppression and ART adherence rates.

### Factors associated with viral suppression

This study explored the Clinical profile of HIV-infected adults receiving a holistic approach of care model in Kampala District. Findings from our study revealed that patients aged 40 to 59 years were more than twice as likely to have viral load suppression as compared to those who were 18 to 39 years. Our findings resonate with those reported before in Uganda [17,18], that older patients were more likely to achieve viral suppression. This can be explained by the fact that the holistic approach was more tailored to the older patients as compared to the younger age groups. Other studies, in Africa, have also reported younger age being linked to unsuppressed viral load [19,20].

Adherence was significantly associated with viral suppression. PLHIV who had good adherence were more than 12 times as likely to have viral suppression. These findings are in tandem with other studies [21,22] which documented that poor adherence is associated with viral load

**Table 3. Bivariate logistic regression for the association between socio-demographic factors and viral load suppression.**

| Variable | Category | cOR | 95%CI | P-value |
|---|---|---|---|---|
| **Sex** | Female | Ref | | |
| | Male | 1.01 | 0.34–2.89 | 0.989 |
| **Age** | 18–39 years | Ref | | |
| | 40–59 years | 2.71 | 1.27–5.78 | **0.010** |
| | 60 years and above | 2.86 | 0.51–16.21 | 0.234 |
| **Prevention of Mother-to-Child Transmission (PMTCT)** | No | Ref | | |
| | Yes | 0.74 | 0.28–1.94 | 0.540 |

Ref- Reference category; cOR- Crude Odds Ratio; CI- Confidence Interval.

non-suppression. This may be explained by the fact that when the drug's concentration is not optimum due to poor adherence, the virus replicates and the viral load does not get suppressed. It may also be that the most widely used ART regimens in the past (2015–2018) included azidothymidine (AZT), Nevirapine (NVP), and Efavirenz (EFV), which were linked to higher rates of ART drug resistance as well as side effects. This may have affected adherence, resulting in failure to suppress viral load.

## Strengths of the study

The study used a large sample size (n = 910) to attain statistical power to meet the research objectives. Data was extracted from different health facilities in the division to ensure variability in the data. Additionally, all field team members were well trained to minimize errors during data collection.

## Limitations of the study

The study was reliant on routinely collected data that was not originally designed for research purposes, hence some data such as marital status, education level, alcohol and drug abuse required for the study was found lacking.

## Conclusion

A holistic approach of care model ensures that patients are empowered in all facets of their lives, as well as fostering the creation of strong communities that can support anti-AIDS efforts

**Table 4. Bivariate logistic regression for the association between clinical factors and viral load suppression.**

| Variable | Category | cOR | 95%CI | P-value |
|---|---|---|---|---|
| **Body Mass Index (BMI)** | Underweight | Ref | | |
| | Normal weight | 1.58 | 0.44–5.73 | 0.483 |
| | Over weight | 1.20 | 0.43–3.35 | 0.727 |
| | Obese | 0.77 | 0.27–2.18 | 0.622 |
| **HIV regimen** | TDF-3TC-EFV | Ref | | |
| | AZT-3TC-NVP | 0.88 | 0.53–1.46 | 0.627 |
| | Other regimen | 1.09 | 0.35–3.43 | 0.876 |
| **Adherence** | Not good | Ref | | |
| | Good | 10.52 | 1.61–68.66 | **0.014** |

Ref- Reference category; cOR- Crude Odds Ratio; CI- Confidence Interval.

**Table 5. Multivariate logistic regression to determine factors associated with viral load suppression.**

| Variable | Category | aOR | 95%CI | P-value |
|---|---|---|---|---|
| **Age** | 18–39 years | Ref | | |
| | 40–59 years | 2.85 | 1.36–5.97 | **0.005** |
| | 60 years and above | 2.96 | 0.48–18.42 | 0.244 |
| **Body Mass Index (BMI)** | Underweight | Ref | | |
| | Normal weight | 1.84 | 0.53–6.37 | 0.337 |
| | Over weight | 1.35 | 0.53–3.42 | 0.531 |
| | Obese | 0.88 | 0.31–2.45 | 0.802 |
| **Adherence** | Not good | Ref | | |
| | Good | 12.29 | 1.86–81.07 | **0.009** |

Ref- Reference category; aOR- Adjusted Odds Ratio; CI- Confidence Interval; Good Adherence—90% or greater of doses taken as prescribed.

that result in improved treatment outcomes. Our findings show that age and adherence are linked to viral load non-suppression among HIV-infected adults receiving a holistic approach of care model. We recommend for scale up of holistic approaches with an emphasis on the younger individuals living with HIV/AIDS.

## Recommendations

Counselors in ART clinics should offer client-centered HIV therapy to all patients, with a focus on adherence to ART.

Patients between the ages of 40 and 59 seem to be more experienced, so services that recognize peer influencers should target patients in this age group to serve as peer counselors to other PLHIV. The holistic model should accommodate more and specific designs for young people living with HIV/AIDS.

## Supporting information

**S1 File.**
(DTA)

## Acknowledgments

We wish to acknowledge the management of Reach Out Mbuya for their efforts and support during the study.

## Author Contributions

**Conceptualization:** Emmanuel Sendaula, Michael Buwembo.

**Formal analysis:** Emmanuel Sendaula, Denis Opio.

**Investigation:** Emmanuel Sendaula.

**Methodology:** Emmanuel Sendaula.

**Validation:** Denis Opio.

**Writing – original draft:** Emmanuel Sendaula, Michael Buwembo, Denis Opio.

**Writing – review & editing:** Emmanuel Sendaula, Michael Buwembo.

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
