## [Decision Letter · Decision Letter 0]

2 Feb 2022

PONE-D-21-11284Clinical profile of HIV-infected adultsreceiving aholistic approach of care model in Nakawa, Kampala District

PLOS ONE

Dear Dr. SENDAULA,

Thank you for submitting your manuscript to PLOS ONE. After careful consideration, we feel that it has merit but does not fully meet PLOS ONE’s publication criteria as it currently stands. Therefore, we invite you to submit a revised version of the manuscript that addresses the points raised during the review process.

The manuscript has been evaluated by two reviewers, and their comments are available below. The reviewers have raised a number of concerns that need attention. They request additional information on methodological aspects of the study, revisions to the statistical analyses and a revision of the English grammar. Could you please revise the manuscript to carefully address the concerns raised?

We look forward to receiving your revised manuscript.

Kind regards,

Elisa Panada

Associate Editor

PLOS ONE

Journal Requirements:

3. Your abstract cannot contain citations. Please only include citations in the body text of the manuscript, and ensure that they remain in ascending numerical order on first mention.

Reviewers' comments:

Reviewer's Responses to Questions

**Comments to the Author**

1. Is the manuscript technically sound, and do the data support the conclusions?

Reviewer #1: Partly

Reviewer #2: Partly

2. Has the statistical analysis been performed appropriately and rigorously? 

Reviewer #1: Yes

Reviewer #2: Yes

3. Have the authors made all data underlying the findings in their manuscript fully available?

Reviewer #1: Yes

Reviewer #2: Yes

4. Is the manuscript presented in an intelligible fashion and written in standard English?

Reviewer #1: Yes

Reviewer #2: No

5. Review Comments to the Author

Reviewer #1: Thanks for the opportunity to review this work on HIV and person-centred care.This is a well written manuscript however, this manuscript will be much clearer to readers if the following points are reviewed and modified:

1. Line 46: Could the author clarify what the 2020 mile stone is about?

2. Line 58 and 63 appeared repetitive, could the author revise this without the repetitions?

3. The sentence towards the end of line 73-75 seems incomplete although it read like the aim/objective of the study, therefore author should review this sentence as appropriate.

4. Line 80-81 Could the author describe what a holistic approach of care model is? What specifically were the patients receiving or what are the care components?

5. Line 104-105 Could the author state the ethics approval reference numbers for the 2 ethics bodies?

6. Line 158 could the author clarify if they are writing viral load or viral road?

7. Line 162 Could the author stick to either 'percent' or '%' for consistency

8. Line 167-169 Could the author revise this statement about 'viral load non-suppression' its a bit confusing to say 'reduced the likelihood of viral load non-suppression by 65%'. If the author is reporting a reduction in viral load suppression by 65%, then state that, and avoid the term 'non-suppression'. This has been mentioned a few times throughout the manuscript, please review and revise.

9. Line 180-181 Could the author define the abbreviations used

10. Line 183-186 Could the author revise the strengths of this study, as this cannot only focus on using a 'well-trained professionals' to collect data.

11. Line 194-196 This final statement is not quite clear...what is the relationship between age, adherence and the use of holistic approaches?

Reviewer #2: The authors reported the clinical profiles of PLHUV receiving holistic care in Nakawa. There are some comments and questions below

Abstract

1. The reference citation is not needed in the abstract

2. The numbers should not be used for the beginning of the sentences

3. The authors said: 84.5% had VL detectable, and 95.3% had UD VL. Which number is corrected?

4. The authors concluded that holistic care had good effects. However, there is no data on comparing "not holistic care."

Text

1. The reference citation is not needed in the abstract. Thus, it should start with 1 in the main text section

2. UNAIDS data is needed to be updated

3. What is the definition of holistic care/system? Please explain in detail and what is the difference from the standard/routine care in the method section

4. The study should be a retrospective cohort study rather than a cross-sectional study

5. Why do the authors use stratified random sampling? Does this mean that all areas there use a holistic system?

6. If possible, systemic randomized should not be used due to the bias. Please explain why using this sampling method

7. What is the statistical program used in this study?

8. Tables should be moved to the back, and more texts are needed to describe the details of the results

9. The abbreviation should be stated under each table

10. Table 4, what is the definition of good adherence?

11. Table 3 and 4, for continuous variables, please try to analyze as a continuous variable and compare the results

12. Table 4, why use the outcome of VL non-suppression. But, the authors discuss VL suppression.

13. Please check references: order and format (e.g., authors list and name of the journal)

Others

1. English language is needed to be correct

2. More details in main methods, results, and discuss section are needed

6. PLOS authors have the option to publish the peer review history of their article (what does this mean?). If published, this will include your full peer review and any attached files.

Reviewer #1: **Yes: **Mary Abboah-Offei

Reviewer #2: No

---

## [Author Response · Author response to Decision Letter 0]

11 Apr 2022

Comment: Is the manuscript technically sound, and do the data support the conclusions?

Reviewer #1: Partly

Reviewer #2: Partly

Response: N/A

Comment: Has the statistical analysis been performed appropriately and rigorously?

Reviewer #1: Yes

Reviewer #2: Yes

Response: N/A

Comment: Have the authors made all data underlying the findings in their manuscript fully available?

Reviewer #1: Yes

Reviewer #2: Yes

Response: N/A

Comment: Is the manuscript presented in an intelligible fashion and written in standard English?

Reviewer #1: Yes

Reviewer #2: No

Response: N/A

Review Comments to the Author

Comment: Line 46: Could the author clarify what the 2020-mile stone is about?

Response: The statement was out of recent context; the line has been revised.

Comment: Line 58 and 63 appeared repetitive, could the author revise this without the repetitions?

Response: Both lines have been revised to give a clear meaning.

Comment: The sentence towards the end of line 73-75 seems incomplete although it read like the aim/objective of the study, therefore author should review this sentence as appropriate.

Response: The sentence has been revised

Comment: Line 80-81 Could the author describe what a holistic approach of care model is? What specifically were the patients receiving or what are the care components?

Response: The holistic approach has been described in the introduction section. Line 66 -74

Comment: Line 104-105 Could the author state the ethics approval reference numbers for the 2 ethics bodies?

Response: All ethical approval reference numbers have been updated

Comment: Line 158 could the author clarify if they are writing viral load or viral road?

Response: Typo error corrected

Comment: Line 162 Could the author stick to either 'percent' or '%' for consistency

Response: The manuscript is revised to %

Comment: Line 167-169 Could the author revise this statement about 'viral load non-suppression' its a bit confusing to say 'reduced the likelihood of viral load non-suppression by 65%'. If the author is reporting a reduction in viral load suppression by 65%, then state that, and avoid the term 'non-suppression'. This has been mentioned a few times throughout the manuscript, please review and revise.

Response: The manuscript analysis has been revised to reflect viral load suppression 

Comment: Line 180-181 Could the author define the abbreviations used

Response: The abbreviations used have been defined.

Comment: Line 183-186 Could the author revise the strengths of this study, as this cannot only focus on using a ‘well-trained professional’ to collect data.

Response: Study strengths have been revised.

Comment: Line 194-196 This final statement is not quite clear...what is the relationship between age, adherence and the use of holistic approaches?

Response: N/A

Reviewer #2: The authors reported the clinical profiles of PLHIV receiving holistic care in Nakawa. There are some comments and questions below

Abstract

Comment: The reference citation is not needed in the abstract

Response: The abstract has been revived

Comment: The numbers should not be used for the beginning of the sentences

Response: The manuscript has been revised

Comment: The authors said: 84.5% had VL detectable, and 95.3% had UD VL. Which number is corrected?

Response: The statement has been revised

Comment: The authors concluded that holistic care had good effects. However, there is no data on comparing "not holistic care."

Text

Response: The reviewer is right, there was no comparator. This was basically a cross sectional study. Generally cross-sectional studies have that weakness in common

Comment: The reference citation is not needed in the abstract. Thus, it should start with 1 in the main text section

Response: All references have been revised and updated

Comment: UNAIDS data is needed to be updated

Response: Reference has been updated

Comment: What is the definition of holistic care/system? Please explain in detail and what is the difference from the standard/routine care in the method section

Response: The holistic approach has been described in the introduction section. Line 66 -74

Comment: The study should be a retrospective cohort study rather than a cross-sectional study

Response: With several technical consultations, we believe that the study is retrospective 

cross-sectional study rather than a a retrospective cohort.

Comment: Why do the authors use stratified random sampling? Does this mean that all areas there use a holistic system?

Response: Yes, all study facilities use a holistic approach

Comment: If possible, systemic randomized should not be used due to the bias. Please explain why using this sampling method

Response: We acknowledge the weaknesses of systemic randomized however will also weighed the strength and simplicity it could introduce to the study such as eliminating the phenomenon of clustered selection 

Comment: What is the statistical program used in this study?

Response: STATA version 13. 0 mentioned in line 25

Comment: Tables should be moved to the back, and more texts are needed to describe the details of the results

Response: This has been done

Comment: The abbreviation should be stated under each table

Response: This has been done

Comment: Table 4, what is the definition of good adherence?

Response: Defined in line 142

Comment: Table 3 and 4, for continuous variables, please try to analyze as a continuous variable and compare the results

Response: Re-analysis done, Age as a continuous variable destabilized the model fit and other variables are binary or ordinal by nature

Comment: Table 4, why use the outcome of VL non-suppression. But the authors discuss VL suppression.

Response: Re-analysis done, VL suppression is currently the outcome

Comment: Please check references: order and format (e.g., authors list and name of the journal)

Others

Response: References have been revised and reorganized.

Comment: English language is needed to be correct

Response: English language has been corrected

Comment: More details in main methods, results, and discuss section are needed

Response: All sections have been updated.

---

## [Decision Letter · Decision Letter 1]

13 Jul 2022

Clinical profile of HIV-infected adults receiving a holistic approach of care model in Nakawa, Kampala District

PONE-D-21-11284R1

Dear Dr. SENDAULA,

We’re pleased to inform you that your manuscript has been judged scientifically suitable for publication and will be formally accepted for publication once it meets all outstanding technical requirements.

Kind regards,

Carla Pegoraro

Division Editor

PLOS ONE

Additional Editor Comments (optional):

Reviewers' comments:

Reviewer's Responses to Questions

**Comments to the Author**

1. If the authors have adequately addressed your comments raised in a previous round of review and you feel that this manuscript is now acceptable for publication, you may indicate that here to bypass the “Comments to the Author” section, enter your conflict of interest statement in the “Confidential to Editor” section, and submit your "Accept" recommendation.

Reviewer #1: All comments have been addressed

2. Is the manuscript technically sound, and do the data support the conclusions?

Reviewer #1: Yes

3. Has the statistical analysis been performed appropriately and rigorously? 

Reviewer #1: Yes

4. Have the authors made all data underlying the findings in their manuscript fully available?

Reviewer #1: Yes

5. Is the manuscript presented in an intelligible fashion and written in standard English?

Reviewer #1: Yes

6. Review Comments to the Author

Reviewer #1: The author has addressed tall the comments made in the first review therefore there is no further comment.

7. PLOS authors have the option to publish the peer review history of their article (what does this mean?). If published, this will include your full peer review and any attached files.

Reviewer #1: **Yes: **Mary Abboah-Offei

---

## [Editor Report · Acceptance letter]

18 Jul 2022

PONE-D-21-11284R1 

Clinical profile of HIV-infected adults receiving a holistic approach of care model in Nakawa, Kampala District. 

Dear Dr. Sendaula:

I'm pleased to inform you that your manuscript has been deemed suitable for publication in PLOS ONE. Congratulations! Your manuscript is now with our production department. 

Kind regards, 

on behalf of

Dr Carla Pegoraro 

Staff Editor

PLOS ONE